# Geomorphological Landscape Research and Flood Management in a Heavily Modified Tyrrhenian Catchment

**Pietro Piana [1], Francesco Faccini [2,3,*], Fabio Luino [3], Guido Paliaga [3], Alessandro Sacchini [4] and Charles Watkins [5]**

[1]  Politics Department, University of Genoa, 16124 Genoa, Italy
[2]  Earth, Environmental and Life Sciences Department, University of Genoa, 16132 Genoa, Italy
[3]  National Research Council, Research Institute for Geo-Hydrological Protection, 10100 Turin, Italy
[4]  Geology and Geography consultant, 16010 Genoa, Italy
[5]  School of Geography, University of Nottingham, NG7 2RD Nottinghamshire, UK
*  Correspondence: faccini@unige.it; Tel.: +39-0103538039

**Abstract:** Since the nineteenth century, most urban catchments in Europe have been subject to significant landscape variations. These modifications have been caused by population change and the transition through rural, industrial and post-industrial economies. Land use and rainfall regime changes, together with land use variations, are frequently associated with flood hazard increase. This paper examines geomorphological landscape changes from the nineteenth century to the present day in the Bisagno Valley, Genoa metropolitan area. The Valley is internationally known for its recurring floods: the last events with fatalities occurred in 2011 and 2014. The extent of landscape change and the history of floods were examined by combining scientific data and information from historical maps, written accounts, topographical drawings and photographs. Historical–geographical and geomorphological analyses were used to reconstruct the runoff for three different periods since 1850. Our results demonstrate that geomorphological landscape variations, including modifications of the river bed, and the abandonment of the countryside and terraces are not sustainable and have progressively allowed an increase in flooding, making it necessary to implement sustainable management policies. In particular, specific spatial urban planning and management measures are necessary in order to mitigate flood hazard and vulnerability.

**Keywords:** geomorphological landscape; land use change; runoff; sustainable landscape planning; Tyrrhenian catchment

## 1. Introduction

Pre-industrial Europe was characterised by an economy based on subsistence agriculture [1], although there was significant trade in agricultural products. Most land was managed for the production of diverse crops, animals and timber and traditional rural practices allowed for sustainable exploitation [2,3]. Depending on physical, ecological and anthropogenic conditions, centuries of landscape management practices led to the establishment and diversification of traditional rural agricultural landscapes in the Mediterranean region [4–6]. Industrialisation and urbanisation were strongly associated in Western Europe with rural depopulation and the marginalisation of rural areas [7]. In Italy, the abandonment of the countryside was particularly dramatic in mountain areas, where depopulation caused loss of landscape diversity, uncontrolled woodland regeneration and a reduction in the maintenance of the stream beds [8,9]. In Mediterranean countries, the movement of people from rural, marginal areas to urban and industrial centres, often located along the coast

or in valley bottoms, resulted in unplanned urban growth, anthropogenic modifications and the consequent increase in geo-hydrologic risk due to floods, mud-debris flows and landslides [10]. Changes in land management, land use and variations in climate affect hydrology and can increase flood risk. The management of landscape and water are strongly linked in both rural and urban areas, although farmland, if managed, is more tolerant to floods than urban areas [11]. Long-term socioeconomic changes in rural and mountain areas have significantly contributed to the change in behaviour of environmental systems. As in the case of the Sudete Mountains, Poland, the reconstruction of these dynamics is greatly facilitated by both field evidence and the use of historical documents. Following rural depopulation, anthropogenic landforms can either disappear within a few years or last much longer. This is true of abandoned terraces, which can influence morphological processes on slopes and runoff [12]. Indeed, historical data are particularly valuable in urban contexts, where the analysis of past flood dynamics can facilitate the identification of hazardous areas and mitigation strategies in cities [13,14]. Well-structured land use planning is one of the best tools to combat increasing rainfall erosivity, which is on the rise in most catchments around the world [15], and land uses within urban river catchments have to be compatible with characteristics and dynamics of fluvial landscapes.

This paper examines anthropogenic landscape changes and flood hazard in the Bisagno Valley (Liguria, Italy). The valley, which is in the Tyrrhenian catchment, is included within the urban area of Genoa, one of the largest coastal cities in the Mediterranean. Due to its high flood risk, the area has been greatly studied in the last 20 years. Previous research has focussed on hydrogeomorphological and meteohydrological aspects; most of the studies focus on the river terminal stretch and its evolution, but very little has been written on historical anthropogenic landscape changes in the Bisagno Valley at the catchment scale and their relations with flood hazard. A detailed knowledge of past landscape dynamics and the connection between rural and urban areas is of crucial importance for sustainable landscape and urban planning policies, particularly in terms of flood hazard management.

Based on these assumptions, the aim of this work is to reconstruct landscape dynamics from the pre-industrial period to present day, and to establish their connection with floods.

The analysis seeks to track links between the rural and the urban area, by looking at the landscape of the whole catchment. In particular, the paper focuses on the effects of human interference on the landscape in terms of morphological modifications and land use changes, both significantly influenced by the socioeconomic and demographical context. The methodology combines historical–geographical and scientific approaches, exploring a wide range of data including historical cartography, topographical art and written documentation, as well as modern digital terrain information and detailed field data from direct surveys. The research focuses on important landscape features such as terraces and built up surfaces, and on the urbanisation and modification of the riverbed. Terraces are particularly important as deep modifications of natural geomorphological dynamics artificially inducing a reduction of slope gradient and solid transport, [16–18]. It is argued that sustainable landscape management policies for marginal and rural areas aimed at the reduction of flood hazard and vulnerability have to combine a qualitative and quantitative approach. In addition, some realistic guidelines for sustainable landscape management and territorial plans for flood reduction are provided.

## 2. Literature Review

Many case studies in Mediterranean countries highlight an increase in the frequency and violence of floods [19,20] and link flash floods and stream power to both climate change and urban sprawl [21–23]. In the Italian peninsula, flood hazard is particularly common due to geomorphologic and climatic features and its primary effects include damage to buildings, infrastructure and loss of life [24]. This is very evident in the north-western Italian region of Liguria, which has been affected by a significant number of flood events in recent years and it is characterised by widespread geo-hydrological phenomena. Previous research focused on coastal areas, looking at the way in which geomorphic processes due to human interference increased geo-hydrological hazard [25–28]. Other studies concerned mountain contexts either in the Po Valley catchment, often independent from human

interaction [29] and coastal catchments characterised by landscape changes and anthropogenic landforms [30]. Ligurian small catchments in sub-coastal contexts lie in intense-rain high-hazard areas and are intensively urbanised along rivers and mainly in lower coastal zones [28]. Some of them have been hit by flash floods in the past; detailed knowledge of their morphometric and morphological features related to flood and landslide hazard, detailed information on flash flood development times, the degree of urbanization/anthropogenic modification and a list of elements at risk can help prioritize effective mitigation strategies [31].

But most studies have focussed on the city of Genoa (NW Mediterranean), which has been affected by flood events since historical times [32] and particularly in the twentieth and twenty-first centuries, the last events with fatalities having occurred in 2010, 2011 and 2014 [33–36]. Recent research demonstrated the strict connection of soil consumption and ultra-centennial oscillation in the metropolitan area of Genoa [37,38]. Within the area of Genoa, the Bisagno Valley constitutes by far the most significant case study, and it is the object of many international studies due to the combination of increased number of floods, related fatalities and damages. Authors looked at the connection between land use cover and runoff, in particular for the assessment of flash flood risk [39–41]. Some studies analysed specific flood events, demonstrating how they were triggered by a combination of elements where recent urban sprawl of the terminal stretch played a crucial role [34,35,42,43]. Other studies concentrated on critical sites, including the large landslide dam of Prato Casarile, in the medium catchment, which constitutes a threat to the population [44]. Other authors [45] used three case studies in the valley to document how, in the last 200 years, the Bisagno Valley in the city has been heavily modified by landscape changes driven by demographic, social and economic factors. As well as constituting a significant case study in scientific literature, the Bisagno Valley and its management are of primary importance in the political agenda at both local and national level. Recent measures to mitigate flood hazard concerned local structural interventions, including the widening and renovation of the river culvert in the terminal stretch and two discharge canals [46]. Public awareness campaigns have been promoted, and population alert techniques have been adopted using new technologies, but recent flood events show that communication and public information on geo-hydrological hazard can be improved [47].

## 3. Study Area

Genoa is a coastal city with a total population of c. 580,000 inhabitants. The urban area stretches for 30 km along the coast, where most of the population is concentrated. The city is backed by the Alps-Apennines chain, which runs parallel to the coast, and the only flat areas are the riverbeds, particularly the Polcevera (W) and the Bisagno (E) valleys, the latter having a total population of c. 150,000 inhabitants.

Since the Middle Ages, the city power has been based on the port, which is still one of the largest in the Mediterranean in terms of size and freight. Through the centuries, the city expanded in the natural amphitheatre around the port area [48]. Up to the late nineteenth century, the economy of the Bisagno Valley was mainly rural, with most of the agricultural production being sold in Genoa. In the twentieth century, the lower Bisagno Valley was progressively urbanised, and some factories were established, while the upper basin underwent significant rural depopulation with consequent loss of managed rural landscapes [45].

Between 1850 and the Second World War, the new urban areas mostly involved the bottom of the valley, which became almost entirely built up [34]. In the post-war period, new urban surfaces are found mainly on the hills of the lower stretch of the valley. The terminal stretch has been progressively incorporated in the city, while its upper part, NE of Genoa, which is included within the municipalities of Bargagli and Davagna still has some rural characteristics. Its drainage basin covers approximately 95 km$^2$ and the main stream has a total length of 25 km (Figure 1, Table 1).

The alluvial floodplain, completely urbanised, is long and narrow—its maximum width being only 300 m. The terminal stretch, almost 1.5 km between the railway and the mouth, is culverted.

The time of concentration of the Bisagno is very short, approximately 3–4 h [49]: this is due to particular geomorphological and land use conditions and slope average gradients, which is approximately 15–20° (Figure 2) The mean rainfall varies between 1100 and 1400 mm, with peaks in autumn (150–200 mm in October). The climate regime is characterised by long and dry summers, followed by short and intense rainfall events which trigger flash floods, particularly in autumn [50]. The maximum flow rate of the Bisagno, with a return time of 200 years, is estimated at 1300 m³/s, which is almost double the capacity of the current terminal culvert.

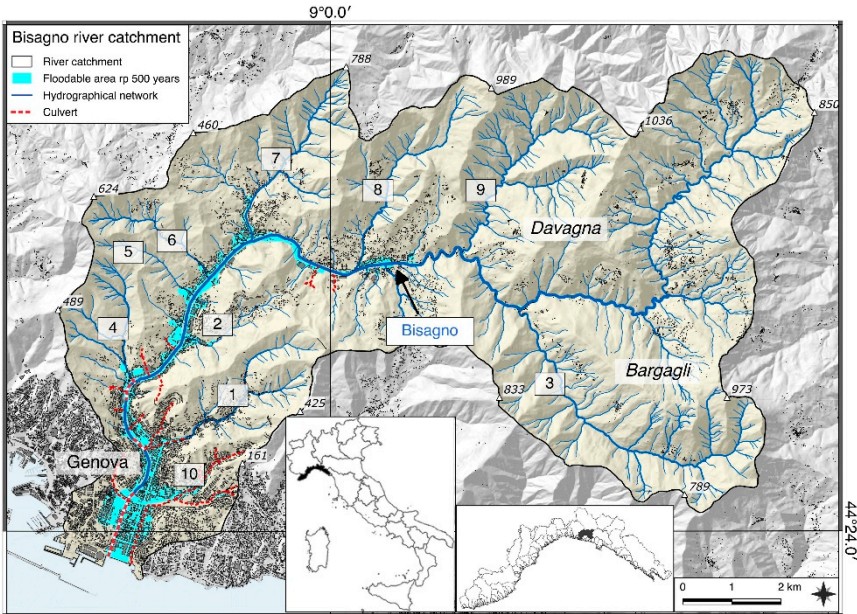

**Figure 1.** Geographic sketch map of Bisagno basin showing the floodplain, completely urbanised, and floodable areas of the valley floor; see Table 1 for details on subcatchments.

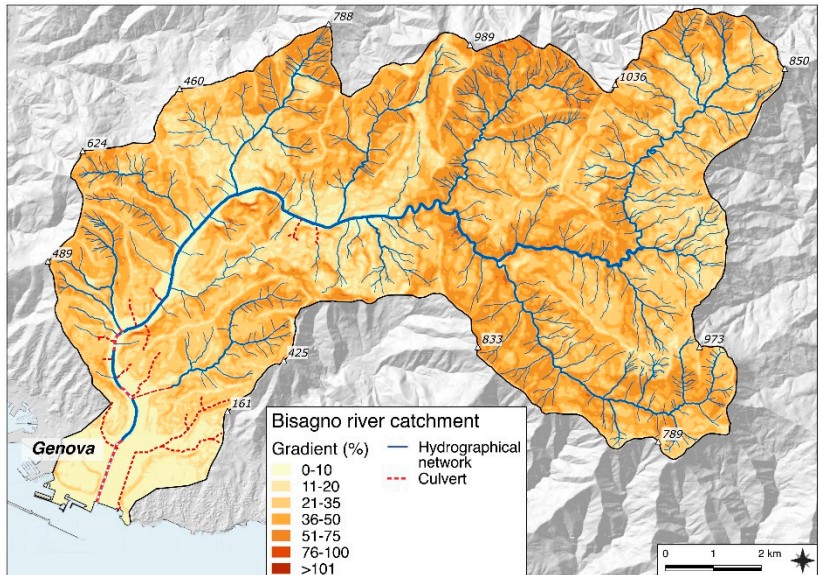

**Figure 2.** Slope map of Bisagno basin, from Digital Terrain Model (5-m resolution); the map shows the main elevations of the basin, concentrated in its northern and eastern part.

**Table 1.** Physical geography features of the Bisagno main tributaries.

| Main Bisagno Tributaries | Slope | Geomorphological Features | Catchment Surface (km²) | Lenght (km) | Max Altitude (M asl) |
|---|---|---|---|---|---|
| 1. Fereggiano | left | Main urban, only upper part natural | 4.9 | 4.7 | 555 (Mt. Forte Ratti) |
| 2. Mermi | left | Main urban | 2.2 | 2.0 | 480 (slopes of Mt. Forte Ratti) |
| 3. Lentro | left | Mountain | 11.6 | 7.5 | 894 (Mt. Becco) |
| 4. Veilino | right | Only plain strongly urbanised | 3.3 | 2.9 | 640 (Mt. Forte FratelloMinore) |
| 5. Cicala | right | Only plain strongly urbanised | 1.5 | 2.5 | 650 (Mt. Forte Fratello Maggiore) |
| 6. Trensasco | right | Only plain strongly urbanised | 2.7 | 3.5 | 670 (Mt. Forte Diamante) |
| 7. Geirato | right | Only plain strongly urbanised | 7.6 | 4.3 | 790 (Mt. Alpe) |
| 8. Torbido | right | Only plain strongly urbanised | 6 | 4.8 | 984 (Mt. Alpesisa) |
| 9. Canate | right | Mountain | 8.8 | 5.4 | 1032 (Mt. Bastia di Marsiglia) |
| 10. Noce | left | Totally urbanised and artificially diverse | Originally c. 3 | 2.7 | 320 (Camaldoli) |
| Bisagno | | | 95 | 25.0 | 1034 (Mt. Candelozzo) |

## 4. Methods

In order to identify and quantify geomorphological landscape change in the Bisagno Valley over the last 200 years and to assess the potential links with flood events, we integrated a historical–geographical approach with a scientific and quantitative methodology. These combined methods allowed us to compare landscape over three centuries and to produce maps of land use in the nineteenth, twentieth and twenty-first centuries alongside a map of typical stone-made terraces and one of anthropogenic landforms [51]. Research in landscape studies increasingly benefit from multidisciplinary approaches and multisource analysis, and the value of landscape history as a planning tool is increasingly recognized [52,53]. Simplified hydro-geomorphological analyses provided information on runoff. The different maps, both historical and modern, used in this work are summarised in Table 2. Some guidelines on flood management in urban planning policies are discussed and developed.

**Table 2.** Overview of the maps used in the research.

| Map | Authority | Scale | Year | Fig./Tab. Reference |
|---|---|---|---|---|
| Printed topographical map | Savoy Kingdom Army | 1:50,000 | 1853 | 3a, 4, 5 |
| Various manuscript topographical maps | Savoy Kingdom Army | 1:9450 | c.1820 | 5, 10a |
| Topographical map | Italian Military Geographical Institute | 1:25,000 | 1907, 1939 | 7 |
| Topographical map | Liguria Region | 1:25,000 | 1995 | 7 |
| Thematic maps related to regional land use | Liguria Region | 1:25,000 | 1973–1975 | 1, 2, 3b, 4 |
| Technical map | Liguria Region | 1:5000 | 1990–2007 | 4, 10b–e, tab1 |
| Digital Terrain Model | Liguria Region | 1:5000 | 2007 | 1, 2, 7, 10b–e |
| Land use | Liguria Region | 1:25,000 | 2012 | 3c, 5 |
| Urban evolution | Liguria Region | 1:25,000 | 1986 | 7 |

### 4.1. Land Use Change Assessment

Landscape variations have been examined using historical maps, topographical views and historical photographs. We reconstructed land use for three dates: 1850 (before industrialization); 1970 (at the height of the industrial boom); present-day (in the post-industrial period). The 1850 sketch map provides information on six categories of land use (woodland, fields, pasture land, 'barren' land

(*Gerbido*), vineyard, and olive orchard); a seventh category, urban area, was added. Information on land use in the nineteenth century was combined with data from other contemporary documents such as other more detailed maps, drawings, historical photographs.

In order to compare data from different periods and with different levels of detail, the categorisation was simplified [54,55]. The *Gerbido* category includes unproductive areas characterised by rocky ground with Mediterranean shrubs or moorland [56]. The definition is typical of a classification system based on production and differs from modern cartographic categories; for this research, these areas are called 'bare land' making it possible to compare nineteenth-century and modern maps. Liguria is world-famous for its terraces sustained by dry stone walls, which characterise the landscape but, at the same time, modify slope stability and dynamics [57,58]: the map of terraces was produced using existing cartography, scientific literature [16] novel field surveys and Google Earth Pro.

In addition, topographical drawings and paintings by local artists and historical photographs were analysed and compared to modern images in order to reconstruct the landscape history of some sites [59,60]. These were particularly valuable in documenting changes to the terminal stretch of the valley. Further archival and bibliographical research provided information on the history of floods in the Bisagno in the last 200 years [32,45]. Flood events were reconstructed in terms of rainfall intensity and ground effects. For the five most significant floods of the last two centuries, maps of flooded areas in relation to urban surface were produced.

### 4.2. Scientific and Quantitative Evaluation

The effects of landscape changes and urban sprawl were analysed in terms of runoff, concentration times and flooded areas. In order to link runoff and concentration time to land use change over two centuries, the non-deterministic Curve Number (CN) model, developed by the Soil Conservation Service was used [61]. This method is based on the CN parameter, which defines the infiltration; the specific saturation volume depends on the lithological and pedological characteristics of the soil and on land use. The soil's tendency to contribute to the discharge depends on three main factors: permeability of soils and underlying bedrock; vegetation and land use; soil saturation [62].

The CN, variable between 0 and 100, defines the soil conditions from a runoff point of view based on the humidity conditions registered for five days before a flood event [63]. Values of approximately 0 indicate a completely permeable surface, while values of approximately 100 points indicate impermeable terrains where the rainfall becomes runoff.

Finally, data of flooded areas reconstructed from scientific reports, newspapers, photographs and videos allowed us to produce a map comparing the flood events in 1822, 1970, 1992, 2011 and 2014.

## 5. Results

### 5.1. Landscape Changes

#### 5.1.1. Land Use Changes

The land use of the Bisagno catchment for 1850, 1970 and 2018 is shown in Figure 3. A remarkable decrease in agrarian surfaces is directly proportional to the increase in urban areas, which increased from 7.9 to 17% of the catchment. Fields and vineyards decreased dramatically, particularly the latter, which dropped down to 0. In some areas, vineyards were replaced by olive plantations, which is the only stable land use class. In 1850, cultivated fields were spread across the whole basin, while in 1970, they were concentrated along the bottom of the valley and at lower elevations. The higher slopes, which in the nineteenth century were mainly open and used as pastureland and for haymaking, went through progressive secondary reforestation and, today, woodland is the most prominent land use category, covering over half of the entire basin.

The map of Figure 4 shows the total surface of terraced areas in the Bisagno catchment (30%) in relation to the Genoa urban area and rural settlements in 1850, highlighting: (a) the distribution of

terraces around the villages; (b) how, today, managed or cultivated terraces only represent 26% of the total terraced area. In accordance with the results of Figure 3, which shows the progressive loss of cultivated fields and vineyards in the rural parts of the valley, the reduction of managed terraces is more significant in the upper catchment, distant from the main settlements.

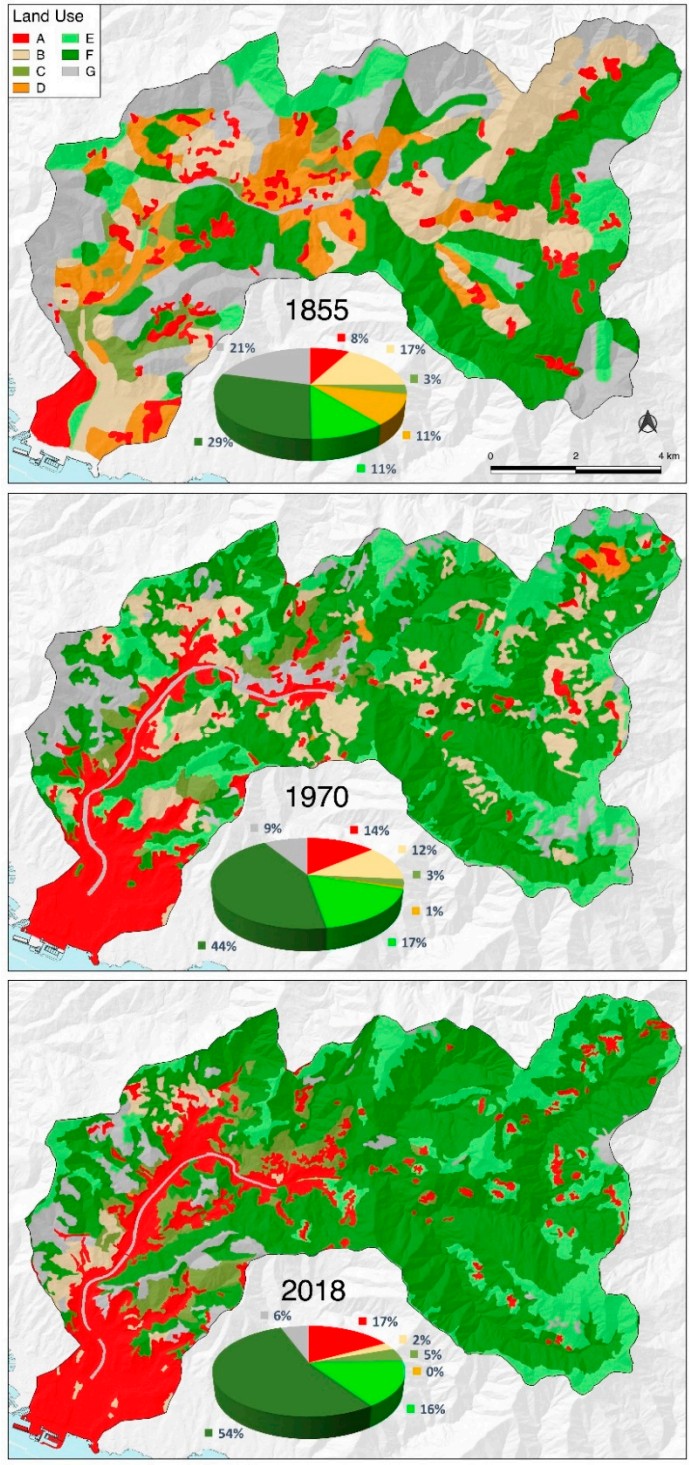

**Figure 3.** Land use sketch map of the Bisagno catchment in 1855, 1970 and present-day. A: urban, B: field, C: olive orchard, D: vineyard, E: pastureland, meadow, F: wood, and G: bare land, rock. The map underlines the high landscape diversity of the nineteenth century compared to 1970 and 2018. Today, two colours prevail, showing the significant extent of wooded surface and urban area.

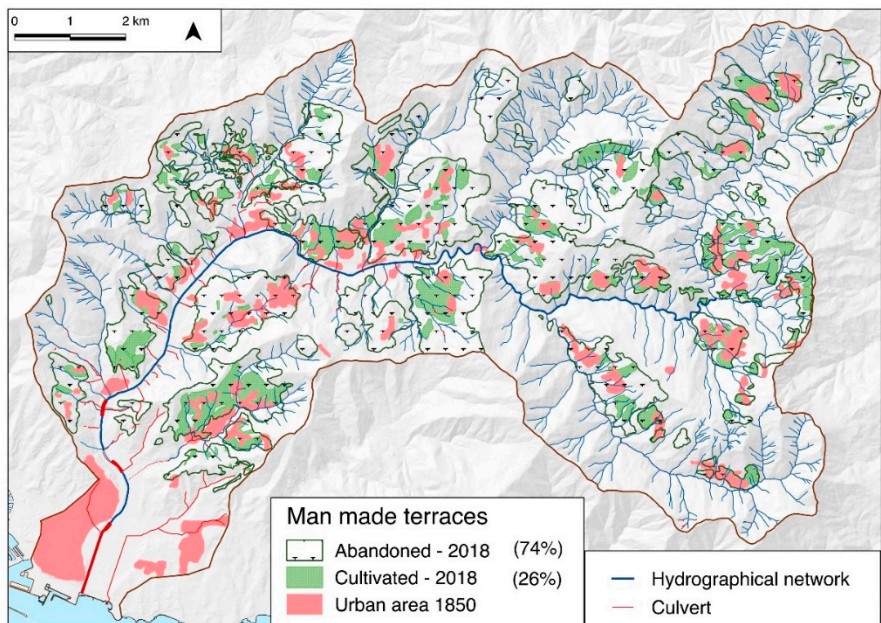

**Figure 4.** Terraces with dry stone walls map. Abandoned terraces are mainly located far from the urban area.

For the analysis of the Curve Number, we applied the parameters from the SCS-CN tables [61,64] assigning different values to two categories, woodland and urban surface, depending on the year. If compared to 1850, the urban surface, today, is more compact and continuous, particularly because of the presence of a dense network of new asphalt roads (CN = 80–98), while woodlands, today, are in poorer condition due to lack of management (CN = 70–77). The CN passes from 75 in the nineteenth century to 80 in 1970 when most of the landscape variations had occurred. The analyses show an increase in the CN in the last 200 years due to land use changes: the significant CN increase due to urbanisation of the floodplain is only partly balanced by the increase in woodland in the upper catchment (Figure 5). The concentration time passes from 3h 16′ to 2h 48′—a very significant variation in a small basin with very rapid hydrological responses.

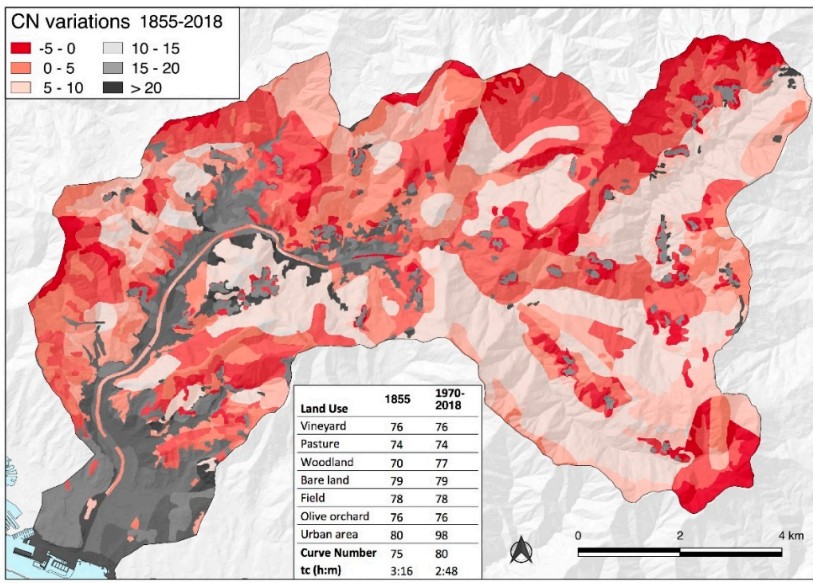

**Figure 5.** Curve Number variations between 1850 and 2015; the table shows its recent increase which is due to land use changes, particularly the increase in woodland and urban area and the reduction of managed agricultural land.

5.1.2. Urban Geomorphology

The comparison between paintings and photographs of different historical periods provides evidence of the remarkable landscape changes in the lower Bisagno valley in the last three centuries (Figure 6). Painting (A) is a view by the Genoese artist Alessandro Magnasco (1667–1749) which depicts the "villa landscape" typical of the Genoese countryside in the pre-industrial period. The painting highlights the regular pattern of walled fields in the valley bottom, which was intensively exploited for agriculture and characterised by scattered houses. The hills around, partly terraced, are completely open and scarcely urbanised, the main settlements being along the ridges. The situation is very similar in 1825—the date of a watercolour by Luigi Garibbo (1784–1869) (Figure 6B). The view is taken from Madonna del Monte, on a prominent hill along the left watershed of the valley which became a very popular viewpoint among artists in the nineteenth century [65]. Garibbo depicts with detail the diverse and well managed rural landscape of the valley in the early nineteenth century, characterised by small cultivated fields, olive orchards and vineyards on terraces, with the city of Genoa, surrounded by the city walls in the background. By 1920, the situation is completely different: these are the years of Genoa's industrial and urban development, and as the photograph of Figure 6C shows, the bottom of the valley is almost completely built up, while the hill above is only partly urbanised. By this period, the lower Bisagno valley had already almost entirely lost its traditional agriculture, but the process soon involved the hills around the city, which by 1970 also became heavily urbanised. Today, both the valley and the hills immediately above are intensively built up, with only the highest part, next to the old Genoa walls, still open. Hardly any sign of rural exploitation and management is detectable in the landscape.

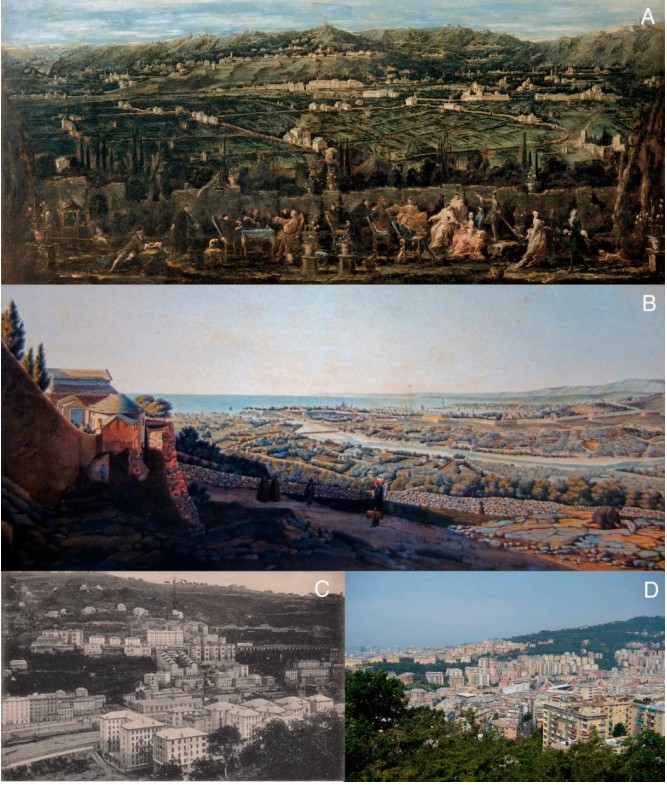

**Figure 6.** Historical images and landscape changes in the lower stretch of the Bisagno Valley: (**A**) Alessandro Magnasco, 1740 (collezione Palazzo Bianco, Genova, from https://bit.ly/33Qw4fZ); (**B**) Luigi Garibbo, c. 1825 (Centro DocSAI, Genova, Coll. Topografica); (**C**) historical photograph, 1929 (from http://ceraunavoltagenova. blogspot.com/); (**D**) present-day. The views, taken from the eastern side of the valley, document the landscape of the pre-industrial period, when the Bisagno area was devoted to agricultural production for the city of Genoa. The urbanization process started in the early twentieth century, initially along the valley and successively on the hills. Today, urbanization is complete.

Figure 7 shows the main anthropogenic landforms of the Bisagno stream catchment, an area internationally known for its artificial grounds that have been a dominating morphogenetic process since the C19th. Rosenbaum et al. [66] classifies them into made ground, worked ground, infilled ground and landscaped ground. an additional category is represented by main and secondary hydrographical network. Amongst made grounds, we can summarise dumps, landfills, sea embankments, railway and motorway. Worked grounds are open quarries for the extraction of marly limestone, whose unmanaged fronts often undermined slope stability. Amongst infilled grounds, there are mid-slope roads, with cuttings and fills and some quarries, partly refilled today. Terraces are a particular example of landscaped ground; their building, which probably dates back to the Middle Ages [16], means significant modifications of the slopes of the most superficial stratigraphy of the soil. Built up areas are landscaped grounds too, both along the valley and on the slopes; their buildings entailed a large movement of material, although in this case the distinction between areas of excavated and areas of made ground is difficult to discern. The main hydrographical network has been intensively modified, with culverts, canalisations and narrowing of the river bed, while minor streams are characterised by river bridles built to reduce erosion and improve embankment stability. The valley is crossed by a railway track and by a motorway, while a mid-slope narrow-gauge railway goes along the right-hand side of the valley.

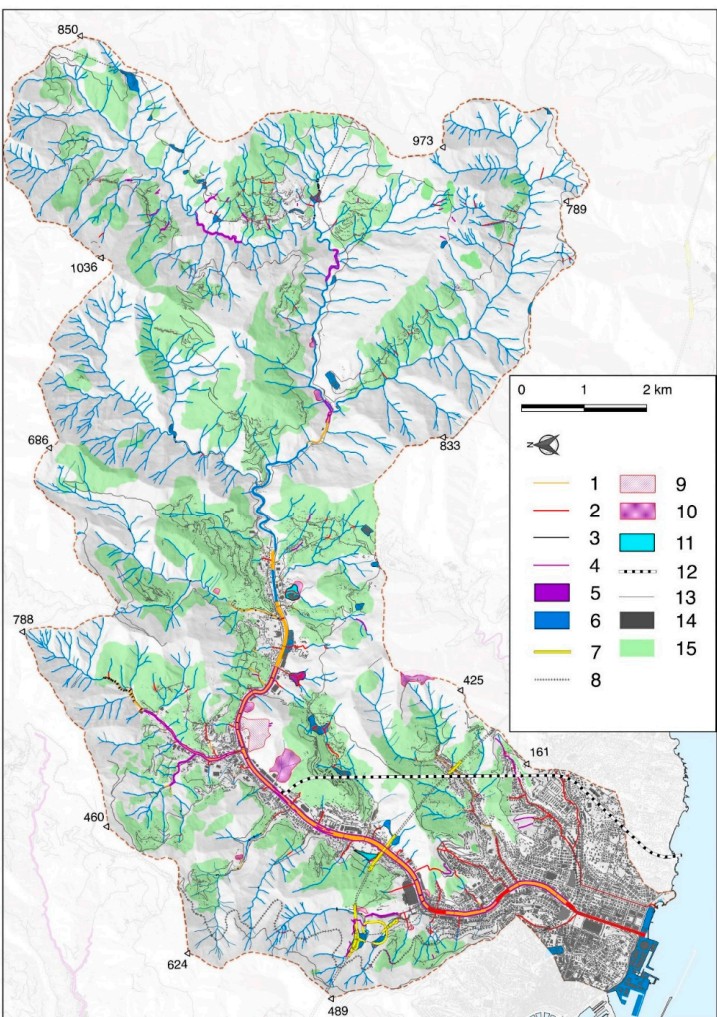

**Figure 7.** Anthropogenic landforms in the Bisagno stream catchment: Modified river networks: 1. Artificial rivernetwork; 2. Culvert; 3. Weir; 4. Riverbank. Made grounds: 5. Dump; 6. Landfill, embankment; 7. Motorway; 8. Railway; Worked grounds: 9. Active quarry; 10. Inactive quarry; 11. Excavation surface; 12. Drainage channel (final design). Infilled grounds: 13. Road. Landscaped grounds: 14. Buildings; 15. Terraces (with dry stone walls).

*5.2. Floodplain Changes and Main Floods*

5.2.1. Floodplain and Riverbed Changes

The Bisagno riverbed channel has been severely constricted in the last 200 years. Figure 8 focuses on the terminal stretch of the valley, providing evidence of the progressive narrowing of the riverbed alongside urbanisation. The process of urbanization of the Bisagno floodplain started in the first decades of the twentieth century and, today, the area is almost completely built-up. The first drawing, by the Genoese artist Luigi Garibbo shows the situation in the early nineteenth century, with the wide and empty river bed of the Bisagno crossed by the St. Agata and Pila bridges; before the narrowing of the river channel, they were respectively 280 and 120 m wide. The floodplain had several water channels used for irrigation and water regulation. Until 1823, the village of Borgo Pila, next to the river mouth, was five metres higher than today; it was lowered at the end of the nineteenth century to facilitate the road link with a new road to the centre of Genoa (now Via XX Settembre) over Pila Bridge. In addition, a small hill between Borgo Pila and the city centre, which was part of the river's western edge, was dismantled in the same period. The situation is similar in the mid-nineteenth century, when the British photographer Jane St John (1801–1882) took a photograph of the area from the old city walls (*FrontiBasse*); the riverbed is wide and open, and the landscape has only scattered buildings and settlements. Today, urbanisation is complete—the embankments are characterised by roads, warehouses and factories while the alluvial plain and the sides of the valley are heavily built up with residential buildings and more culverts were established. The bridge of St. Agata is no longer visible from the same viewpoint; it is 70 m wide and only 5 arches remain of the original 28, while the Pila Bridge no longer exists.

Figure 9 documents the gradual occupation of the fluvial plain as shown in historical maps from four different periods between 1878 and 1997. The buildings have increased from 65,000 to 305,000 from the second half of the nineteenth century to today [67] and the river width decreased dramatically, passing from c. 250 metres (1878) to 50 metres (today) at St. Agata. By 1930, new buildings, roads and railways had been established including the new Genoa–Tuscany railway line and Brignole railway station, and the river was already partly channelled. The maps show the urbanisation of the plain and the construction of the final culvert on the Bisagno in c.1930. However, the flow section was based on incorrect calculations which underestimated the maximum flow rate of the Bisagno, based on the assumption that the maximum flow rate could not exceed 500 m$^3$/s [45]. This value is significantly smaller than the water flow peaks that, in recent years, the stream has reached during its most violent floods. In the same period the Fereggiano Stream (the last left tributary of the Bisagno), was embanked first and then covered for over 1 km. Other parts of this stream were culverted in the 1950s and at the beginning of the twenty-first century. Today, almost 40,000 people live in the terminal stretch of the Bisagno Valley (Foce district), characterised by a very high flood hazard [33].

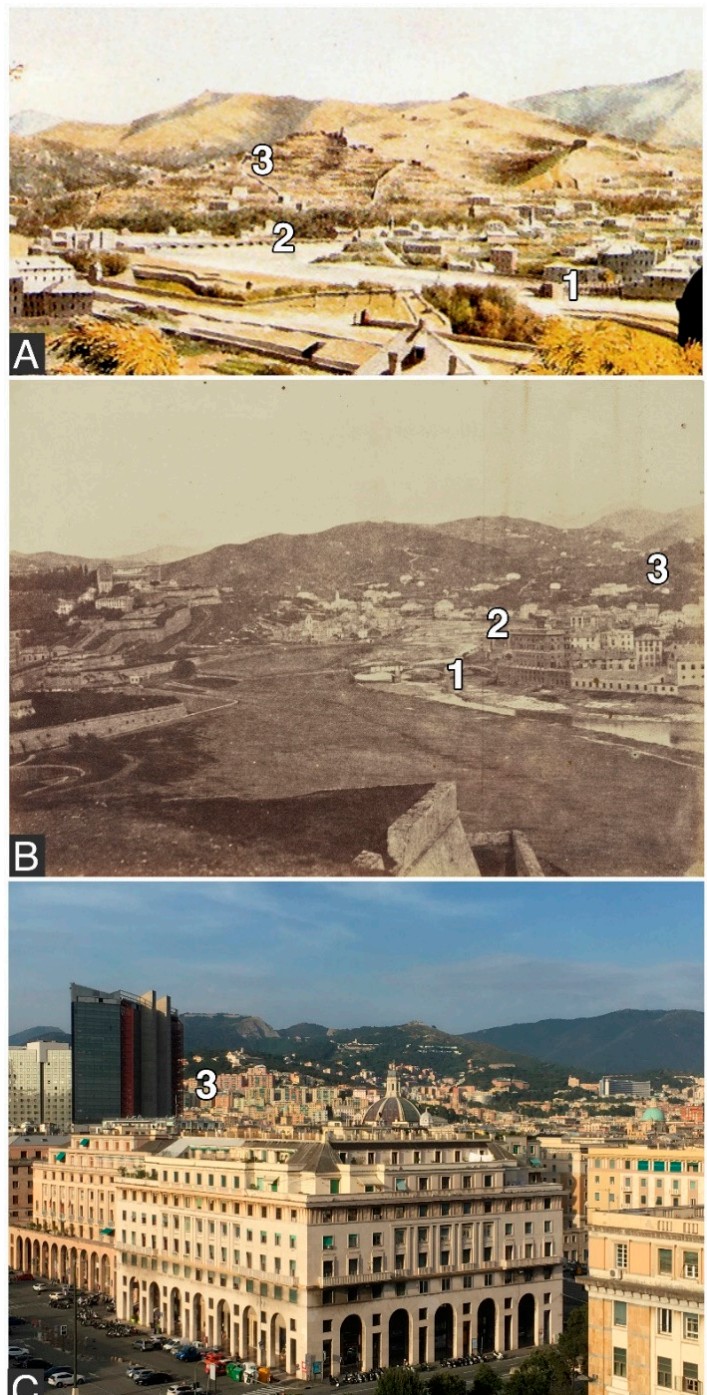

**Figure 8.** Photographical comparison of the final stretch of the Bisagno floodplain from the western side of the valley at Carignano: (**A**) L. Garibbo, 1822 (Centro DocSAI, Genova, Coll. Topografica); (**B**) Jane St John, *Genoa from the Ramparts*, 1856–1859; The J.Paul Getty Museum, Los Angeles, Digital image courtesy of the Getty's Open Content Program; (**C**) present-day. The numbers show three specific sites of the valley. **1:** Pila bridge; **2:** Sant'Agata bridge; **3:** Monte hill.

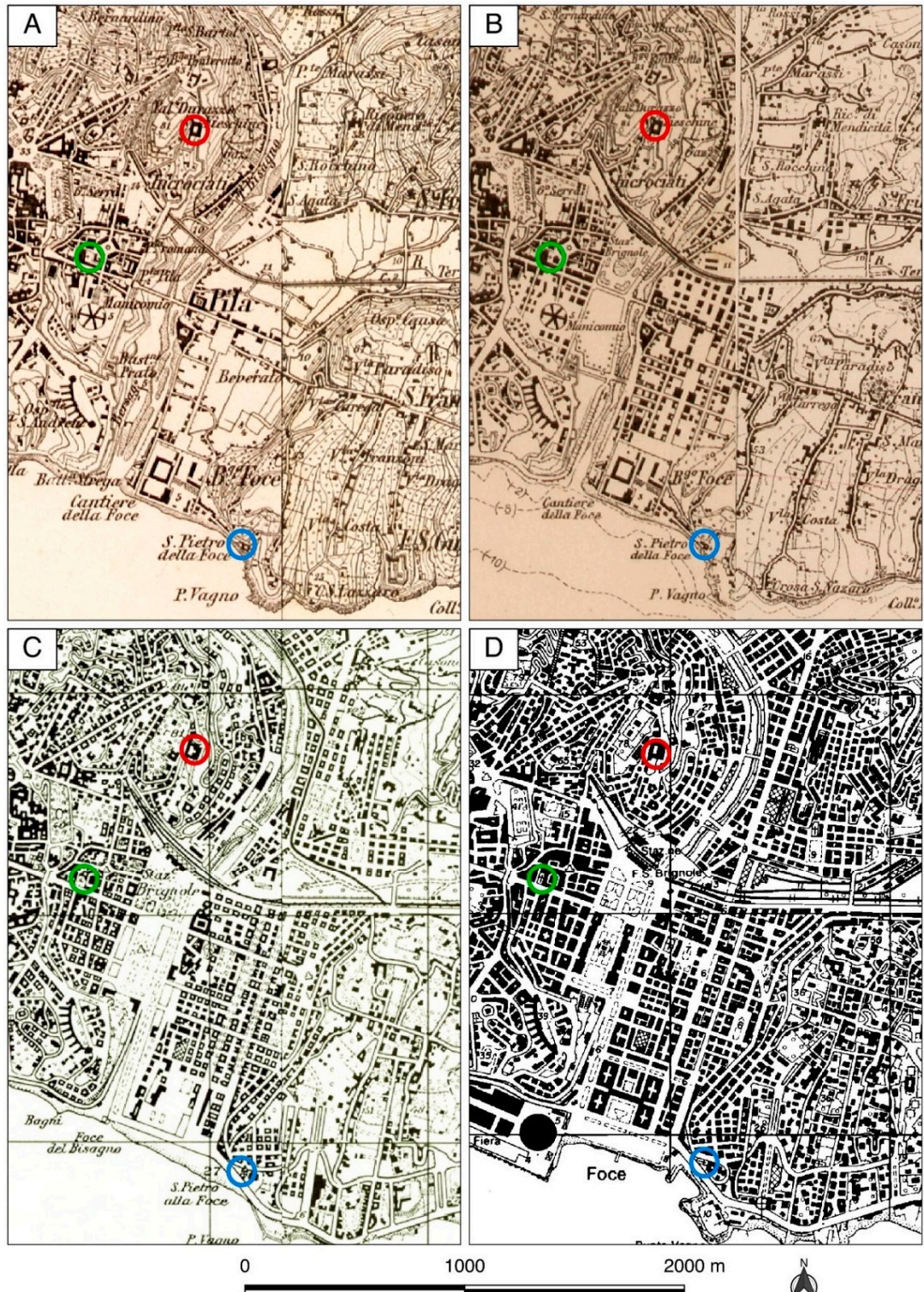

**Figure 9.** Multitemporal comparison of historical maps of the terminal stretch of the Bisagno Valley: (**A**) Tavoletta Istituto Geografico Militare Italiano (IGMI), 1878; (**B**) Tavoletta IGMI, 1907; (**C**) Tavoletta IGMI, 1939; (**D**) Regione Liguria, 1997. **Blue circle**: San Pietro church; **Green circle**: Consolazione church; **Red circle**: Fieschine convent.

### 5.2.2. Main Floods of the Bisagno

Tables 3 and 4 report the main flood events of the Bisagno since the nineteenth century. Four of them occurred in the nineteenth century, six in the twentieth century and two in the early twenty-first

century. The maps of Figure 10 show the flooded area of the five most significant flood events of the period. The terminal floodplain is flooded in every flood event, although with different distribution and water heights depending on the flood entity and the morphological and urbanisation conditions.

The Bisagno plain, not yet urbanised, was completely flooded on 25 October 1822, when contemporary chronicles report '12.5 palms' of water (1 'Genoese palm' = 24.8 cm) and a total amount of rainfall of 30 inches of rain (812 mm) [45]. The flood was very destructive and many buildings, roads and bridges of the Bisagno were damaged, while the number of fatalities is unknown.

The 7–8 October 1970 flood was the most dramatic event of the last two centuries. an exceptional amount of rainfall (948 mm/24 h) was measured in Bolzaneto (Polcevera Valley, north-west of Genoa centre) The maximum flow rate was estimated as 950 m$^3$/s while the maximum flow rate capacity of the Bisagno coverage was less than 700 m$^3$/s. There were 44 fatalities and the estimated cost of the damage was approximately 55 million euros equivalent. The map shows the Bisagno plain downstream of Marassi, completely flooded as in 1822 with a new flooded area westward, directly linked with the significant morphological modifications carried out in the late nineteenth century. The maximum water height was three metres.

On 27 September 1992, two supercell systems struck the Bisagno and Sturla basins at 6 pm and 10 pm: the rainfall measured in Genoa was 421 mm/24 h. The Bisagno stream flooded the plain after the second storm at approximately 12 pm and the flooded area stretched from the Staglieno Monumental Cemetery to Foce at the river mouth. The maximum water height was measured at the mouth of the culvert (1.80 m). Damage was estimated at approximately 75 million euros and there were two fatalities.

The 4 November 2011 event was a flash flood which struck the Fereggiano Valley with a peak of rainfall of over 500 mm/6 h [32]. The stream rapidly overflowed at approximately 1 pm, flooding the roads and washing away cars, buses and people; maximum water height was 1.5 m. There were six fatalities and 150 million euros of damage. The rainfall and discharge peaks were concentrated in the Fereggiano, the tributary of the Bisagno, and the flooded surface shown in Figure 10 is different from the others.

The most recent flood event in Genoa took place on 9 and 10 October 2014. Rainfall peaks in the event were recorded in the Geirato Valley (approximately 140 mm/h, 754 mm/5 days). On the evening of 9 October, disastrous flooding of Bisagno and Fereggiano occurred. The water reached heights up to 2.5 m. There was one fatality and damage costing 300 million euros [31]. The flooded surface shows analogies with the event of 1970.

**Table 3.** Main disaster floods of the Bisagno stream in the C19th. Data from [32].

| Storm Event Date | Flood Event | Discharge Estimated in the Final Stretch | Damage and Fatalities |
|---|---|---|---|
| 1822/10/25 | Regular 24-h flood, with a final 3-h peak (total 812 mm/24 h) in the lower Bisagno catchment (Marassi) | 1200 m$^3$/s | Collapse of two bridges and mud and water up to second floor of houses |
| 1842/8/25 | Regular 10-h flood, flooding of many streams in the final part of the catchment | 600 m$^3$/s | Mud and water up to second floor of houses |
| 1872/10/17 | Regular 48-h flood above all in the western and top part of the catchment and flooding of the final stretch | >300 m$^3$/s | Unknown |
| 1892/10/08 | Regular 24-h flood in the upper catchment | >300 m$^3$/s | Unknown |

**Table 4.** Main disaster floods of the Bisagno stream in the twentieth and twenty-first century. Data from [32].

| Storm Event Date | Flood Event | Discharge Estimated in the Final Stretch | Damage and Fatalities |
|---|---|---|---|
| 1907/10/10 | Regular 24-h flood, overflowing in the middle and final stretch of the stream | 500 m$^3$/s | Unknown |
| 1908/07/18 | Regular 9-h flood in the middle-upper part of the catchment, with a 1.5-h rainfall peak | Approximately 450 m$^3$/s | Flooding of the Foce district |
| 1945/10/29 | Flash flood (6 h) in the western slope with overflowing of the final stretch. | >450 m$^3$/s | The river overflowed upstream of the cover |

**Table 4.** *Cont.*

| Storm Event Date | Flood Event | Discharge Estimated in the Final Stretch | Damage and Fatalities |
|---|---|---|---|
| 1953/09/19 | Flash flood (5 h peak) in the western slope and in the middle and lower part of the catchment with complete overflowing | 800 m$^3$/s | Damage losses for Genoa area: ITL 50 billion (over EUR 25 million) |
| 1970/10/08 | Regular 24-h flood, overflowing of the middle and final stretch and tributaries. | 950 m$^3$/s | Damage losses: EUR 19 billion, 1000 people homeless, 50,000 people unemployed, and 44 fatalities (25 within Genoa) |
| 1992/09/27 | Regular floods with final 2-h peak, overflowing of the final stretch | 700 m$^3$/s | Damage losses: EUR 125 million, 250 people homeless, and two fatalities |
| 2011/11/04 | Flash flood in the western and final part of the catchment, overflowing of the final stretch and Fereggiano tributary | 700 m$^3$/s, final culvert under pressure | Damage losses: EUR 155 million, 150 people homeless, and six fatalities |
| 2014/10/09 | Regular 48-h flood with a 5-h peak causing overflowing of the final stretch | Approximately 1000 m$^3$/s | Damage losses: EUR 250 million, 250 people homeless, and one fatality |

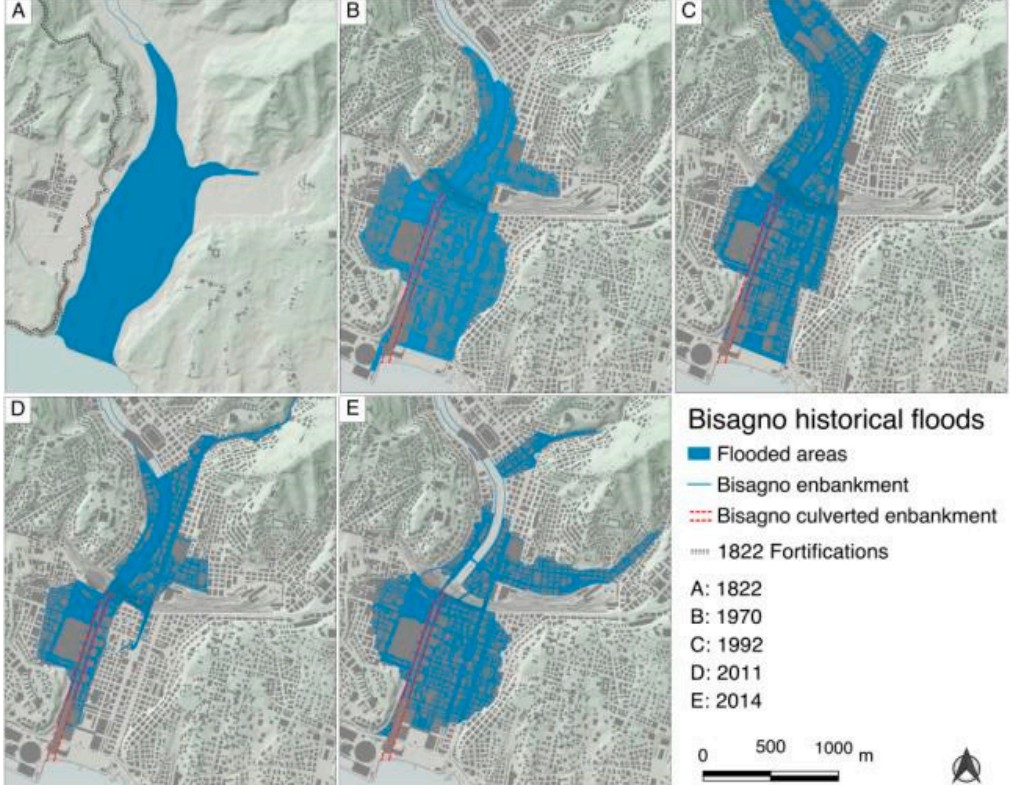

**Figure 10.** Historical flooded areas of the Bisagno floodplain during the most significant flood events of the last two centuries. Since 1970, the Bisagno floods have affected a densely urbanised area.

## 6. Discussion

The aim of this work was a comprehensive reconstruction of landscape dynamics and flood hazard in the Bisagno since the pre-industrial period at the catchment scale. The results demonstrate the importance of a historical–geographical approach to the analysis of land use changes and anthropogenic landforms which increase flood hazard. This is particularly relevant for catchments historically characterised by human activities and urbanisation which have profoundly modified the landscape both in rural and urban areas. This methodological approach, which combines historical documents and current data, can be used for other similar contexts in the Mediterranean. The results allow broader considerations on the effects of anthropogenic changes on flood hazard in the Bisagno Valley and on the way in which sustainable landscape policies can integrate and improve flood management in the area.

### 6.1. Some Remarks on Land Use, Anthropogenic and Demographic Changes in the Bisagno Valley

The documents and maps produced in this paper document significant landscape changes in the Bisagno Valley. Unlike most previous works, which focused on the terminal stretch of the valley, this analysis aimed to reconstruct various features of the valley looking at the whole catchment (Figures 3–5 and 7). It provides evidence of the strict connection and complementarity of two very different areas, the rural and the urban sectors, which went through opposite demographic processes in the last two centuries. Data on population in the Bisagno Valley show that the rural areas of the upper basin in the municipalities of Davagna and Bargagli had an almost constant population between 1861 and the end of the Second World War (Figure 11). With the abandonment of the countryside and the industrialization of the floodplain, there was a dramatic population decline in the countryside and a contemporary peak in Genoa, with over 800,000 inhabitants concentrated in the urban area, including the lower Bisagno Valley. These demographic, social and economic changes profoundly influenced landscape evolution with loss of agricultural and managed surface in the whole catchment, particularly in the upper valley, with the consequent spread of naturally regenerated woodland, similar to other areas of rural Italy (Figure 3, Since the abandonment of the countryside, secondary woodland has progressively covered most of formerly cultivated areas. Lack of management has resulted in infilling of former chestnut orchards with naturally regenerating trees and shrubs and the abandonment of formerly coppiced areas leading to an increase in dead wood. These problems are particularly important in terraced areas, which still occupy 30% of the entire catchment: however, only 26% of the terraces is still managed and cultivated, while the rest is unmanaged and often overgrown with spontaneous vegetation (Figure 4). As other authors argue, when no longer maintained, terraces increase solid transport and they trigger debris flows and shallow landslides [12,68].

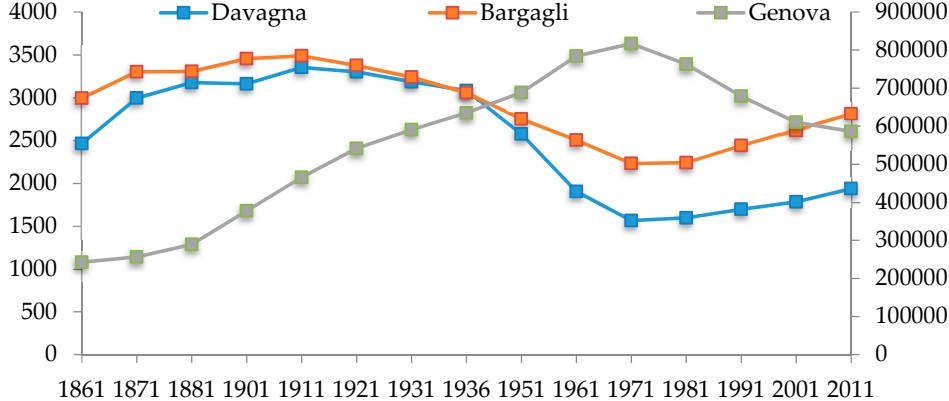

**Figure 11.** Demographic changes in the Bisagno Valley Municipalities in the last 150 years (data from Italian Institute of Statistics).

Following significant demographic increase, today, the floodplain is completely urbanised and the riverbed has been channelled and partly culverted, particularly its final stretch, where most of the floods through the centuries have occurred. In the catchment, the CN has significantly increased, particularly in the floodplain, where most of the population is concentrated. Today, a large number of the population live in highly hazardous areas, particularly the terminal stretch, with a consequent increase in exposed value and significant problems of socioeconomic vulnerability [69].

Anthropogenic landforms characterize the final stretch of the valley floor, but important modifications also involved the valley slopes, as shown by Figure 7. Moreover, a very dense network of paved roads has replaced the old mule tracks, increased soil sealing and cut through the slopes, contributing to soil instability and modifying the hydrographical network.

There is clear evidence that the frequency of floods in the post-war period was higher than in the nineteenth and early twentieth century. The comparison of inundated surfaces for five different flood events showed similar flooded areas (Figure 10). There are, however, substantial differences between the flood of 1822, when the floodplain was not urbanised, and more recent floods. In the nineteenth

century, water could not expand westward to the city centre as the plain was separated from the centre by a hill which was removed at the end of the century. The area around Borgo Pila, along the eastern bank, was excavated and lowered. In addition, inadequate culverts in various stretches of the river play a crucial role [70], and more recent floods affected a larger surface of urbanised area.

Several authors have demonstrated that climate change in Genoa and the Bisagno Valley is already taking place, with negative effects on shallow landslides and flash floods [37,38,40,41,45]. However, an increase in runoff and CN are clearly linked to landscape changes, particularly in the floodplain, where most of the population is concentrated. This analysis provides evidence of how anthropogenic changes, whether direct (urbanization, river channelization, etc.) or indirect (rural abandonment, secondary woodland), are likely to be the main driving factor of the recent increase in flood hazard.

### 6.2. Landscape Planning for Flood Management

Today, flood events in the Genoa area pose a serious threat to public safety and represent a critical problem for local authorities. Flood risk reduction in the Bisagno catchment area is currently one of the most important civil protection objectives in Italy [71,72]. Current structural works include the building of a bypass tunnel, which will carry part of the Bisagno water in case of flood. Other structural interventions have been made to reduce flood hazard. For example, in the western part of Genoa, a building over a riverbed was demolished after the 2010 flood [32]. Non-structural measures have been developed to increase urban resiliency: the system of weather alerts has been improved through the use of text messages, television and radio, while specific plans of actions are adopted in case of orange and red alerts. These measures certainly help to mitigate flood-related hazard, particularly in the light of recent rainfall data (Tables 3 and 4) but the development of landscape and urban planning policies is essential [73].

This paper demonstrated that the rural and urban components of the catchment are directly connected and a comprehensive approach to the area which consider landscape history and anthropic changes is crucial [28]. In the light of the results discussed in this paper, the following guidelines can help in improving current management policies:

(a) **Landscape maintenance:** Small, timely and widespread maintenance interventions, particularly in the upper slope, would reduce runoff and solid transport and increase time of concentration. Given the significant reduction in the time of concentration that has occurred in the last three centuries (Figure 5), widespread interventions are almost crucial, in particular the management of terraces to avoid gullying. Access to funding should be facilitated, while specific financial rewards should be given to local stakeholders involved in sustainable landscape management policies [11]. Land use planning at the watershed level should be promoted and an authority at this level is recommended [74,75].

(b) **Socio-environmental policy:** Following the industrial boom of the post-war period, rural areas became marginal and have been largely neglected by politicians and the public opinion. In this sense, specific policies to encourage rural population recovery and promote sustainable but profitable agriculture and outdoor activities should be developed. The delocalization of businesses, as well as the improvement of transport and infrastructure, could encourage people to move to the countryside and reduce human pressure on the city centre. New sustainable outdoor rural activities in areas which have been devoted to agriculture for centuries would allow a diversification of the land use typology, for the benefit of flood hazard mitigation, as well as for the aesthetic value of the landscape [76–78].

(c) **Landscape planning:** Landscape diversification would also be improved by the establishment of new green urban areas, parks and sustainable urban drainage in the lower Bisagno valley, intensively urbanised and characterized by a very high CN. Green infrastructures would not only reduce runoff, but also contribute to absorb carbon and increase environmental quality near urban areas [79].

(d) **Public engagement:** Urban resilience and public awareness campaigns should be further promoted, and the local population should be actively involved in the management of the territory, including the maintenance of traditional drainage channels associated with terrace agriculture [80,81]. The wealth of historical documents, some of them analysed in this work, are a valuable tool for educational activities on the landscape evolution of the valley and its connection with floods. Further efforts should be aimed at reducing the gap between institutions at national and European level and the local communities, in order to increase their involvement in the decision-making process, a problem which is common to many rural areas of Europe [82].

## 7. Conclusions

This paper considered landscape changes in the Bisagno Valley over the last 200 years in relation to the occurrence of flood events. In the mid-nineteenth century, before industrialisation, the valley was intensively managed for agriculture with terraces covering 30% of the catchment surface. With the industrialisation of the early twentieth century, the economy of the valley went through significant changes as people left the countryside to move to the city. The lower valley became intensively built up and the river bed was channelled and culverted; in the upper valley, terraces and agricultural land were abandoned and new woodland grew over the fields. At the same time, many anthropogenic landforms affected the whole basin, influencing slope stability and increasing soil sealing. The analysis of floods in the last 200 years shows that the area has been historically affected by such events, even in the pre-industrial period. However, since the post-war period, floods have become more frequent and destructive, showing a clear link with landscape changes. The CN has significantly increased, particularly in the lower valley, where most of the population is concentrated.

In order to mitigate flood vulnerability in the valley, specific measures should include broad landscape and urban planning policies aimed to encourage and reinforce agriculture and sustainable land management in the upper Bisagno catchment. This is essential to guarantee timely and adequate interventions at a local level and prevent more significant consequences in terms of geo-hydrological instability. At the same time, structural works in the lower catchment are necessary, particularly the establishment of green areas and sustainable urban drainage. This would increase the landscape diversity in the area and, at the same time, help to reduce the CN. But the most important decision should be the implementation of specific land use planning measurements and regulations aimed to avoid hazard conditions for inhabitants and their proprieties. This research has shown that the combination of methods from earth sciences and historical geography can provide useful insights into the way current landscapes could be sustainably managed in continuity with the past.

**Author Contributions:** Conceptualization, P.P.; methodology, A.S., G.P., P.P.; software, G.P.; validation, P.P.; writing—original draft preparation, P.P., A.S, F.F.; writing—review and editing, F.L., C.W.; visualization, G.P.; supervision, F.F.; funding acquisition, F.F., F.L.

**Funding:** This research was funded in the framework of the RECONECT H2020 project.

**Conflicts of Interest:** The authors declare no conflict of interest.

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
