# Peer review of "Geomorphological Landscape Research and Flood Management in a Heavily Modified Tyrrhenian Catchment"

_sustainability, doi:10.3390/su11174594_

Round 1

Reviewer 1 Report

The paper describes an Italian case study focusing on the connection between
flooding and land use changes within a catchment scale. Some suggestions are
provided below, expecting they are helpful to improve the manuscript
1. Introduction: The manuscript has generally indicated the research background
and a need for research based on sustainable flood risk management. However,
the key research problem and aim are vague. These should be stated more clearly
and directly.
2. Literature Review: There is no specific section to review the-state-of-the-art,
topic-related scholarly articles, books and any other sources. It is important to
provide an overview of flood risk management, landscape change, urbanization
and topic-related issues, and to demonstrate to the audience how your research
fits within a larger field of study.
3. Study Area: The description of the study area needs more citations. Use Figure 1,
2, rather than Figure 1A, 1B.
4. Discussion: The first part of discussion (Lines 346-390) is quite descriptive. It is
more like results, rather than discussion. More importantly, the whole section of
Discussion is rather superficial and descriptive. The argument is not new. From
my point of view, the structure and contents of this section are quite sloppy, with
little attempt to digest and synthesize the information and to derive a coherent
argument.

Author Response

The paper describes an Italian case study focusing on the connection betweenflooding and land use changes within a catchment scale. Some suggestions areprovided below, expecting they are helpful to improve the manuscript

Introduction: The manuscript has generally indicated the research background and a need for research based on sustainable flood risk management. However,the key research problem and aim are vague. These should be stated more clearlyand directly.

Many thanks for your suggestions. We have reviewed and amended the Introduction considerably, stating the aim and objectives and the research questions of the paper considering the importance of the study area both in scientific literature and in the political agenda.

Literature Review: There is no specific section to review the-state-of-the-art, topic-related scholarly articles, books and any other sources. It is important toprovide an overview of flood risk management, landscape change, urbanizationand topic-related issues, and to demonstrate to the audience how your researchfits within a larger field of study.

Within the Introduction section, the initial paragraphs are dedicated to the state of the art. We discuss relevant papers in depth, looking at methodological issues as well as content. We analyse papers dealing with similar case studies in Europe and Italy, before focussing on previous research on the Bisagno Valley, which is one of the most significant case studies internationally

Study Area: The description of the study area needs more citations. Use Figure 1,
2, rather than Figure 1A, 1B.

We have added more citations in the Study Area section and we have divided the two figures, which are now Figure 1 and Figure 2

Discussion: The first part of discussion (Lines 346-390) is quite descriptive. It ismore like results, rather than discussion. More importantly, the whole section ofDiscussion is rather superficial and descriptive. The argument is not new. Frommy point of view, the structure and contents of this section are quite sloppy, withlittle attempt to digest and synthesize the information and to derive a coherentargument.

We have restructured this section and added some new material to strengthen the argument. We have also moved some material  into the Results section. We have divided the Discussion into two separate sections. We have decided to keep Figure 11 (demographic changes in the valley) to stress the clear link between human presence and landscape changes. Given the results explained by maps and pictures of Figure 3-8, we argue that the increased occurrence of floods in the area is mostly caused by human interference, whether direct (urbanisation) or indirect (land abandonment). We discuss issues of exposed value and vulnerability due to the intense urbanisation of the riverbed, documented in the results by a new version of Figure 8 and a new figure, Figure 9, which compares 4 historical maps of the final stretch. However, we also suggest that comprehensive landscape and urban planning has to consider rural areas and that the direct involvement of the local population is crucial.

Reviewer 2 Report

Comments to Authors

The presented manuscript analyses the landscape variations caused by strong urbanization in a coastal catchment of northern Italy.

From this point of view, the article is original and significant to warrant publication in Sustainability. Method used and data available justify the interpretations and conclusions.

I recommend publishing this manuscript with minor revisions.

Anyway, I found some inaccuracies in the text and in the figures, which are listed below.

General consideration

There is a problem in the title and all over the manuscript regarding the use of the term risk , which is used in inappropriate way. I don’t have to remember to the Authors that to evaluate risk they have to consider hazard, exposed value and vulnerability. None of these parameters is determined by the Authors so I suggest to avoid speaking of flood risk. The data Authors presented are dealing only with the influence of man induced land use variations on the increase of flooding events.

-          Line 32: add   flooding or flash flood

-          Line 49: in this session the Authors should add papers dealing with this problem in other contexts of Europe and Italy.

-          Line 104: be more precise about concentration time. Add values.

-          Lines 281- 284:   this description needs a more detailed figure than figure 6.  I suggest to enlarge the more urbanised are near to the coast, trying to show the main human modification the natural river network.

-          294: please try to add more information about the reduction of river width

-          402-403: Are you affirming that ”small and widespread interventions should reduce .. time of concentration?  Do you mean increase? Anyway, taking into account rainfall data of Table 3, which show events with very high values of rainfall intensity, mitigation measures have to consider a reduction of the exposed value in association with interventions works in the catchment. The sole change in land use will not be sufficient to risk mitigation.

-          438-442: Is the paper dealing with flooding or with geo-hydrological risk in general ? Please avoid to use this term.

-          382-390:  The urban expansion cannot be the only reason of the increase of flood frequency in the post war period.  What about climate variations (see lines 379-381)? Urban expansion has increased the exposed value, not the flood frequency. There is a lot of confusion, please delete or re-write these lines

Tables and Figures

-          Fig. 1:  add boundaries of sub-catcments listed  in tab 1.; delete admistrative boundaries; ; What does mean the “s “near Bisagno ? stream?; longitude values are not readable. Fig 1b= Are you sure that slope classes are expressed   as degrees? >101°????

-          Tab 1: please add decimal value to river lenght column.

-          Fig 2: Please, outline better the river catchment boundary, the outlet and the coastline in this figure and in the following ones (fig. 3 and 4). Add the symbol % in the diagrams

-          Fig3: add main river network.

-          Fig. 6: see comment to lines 281-284

-          Fig. 7: Please add some targets on the photos to allow comparison

-          Tab.4: add references for rainfall data, and a column for damages and victims with references.
-           Fig.8: Try to express these data as % of the total urban area in each moment.

Author Response

There is a problem in the title and all over the manuscript regarding the use of the term risk , which is used in inappropriate way. I don’t have to remember to the Authors that to evaluate risk they have to consider hazard, exposed value and vulnerability. None of these parameters is determined by the Authors so I suggest to avoid speaking of flood risk. The data Authors presented are dealing only with the influence of man induced land use variations on the increase of flooding events.

Many thanks for your suggestions. Rather than risk, we now talk about flood hazard and vulnerability. We have also changed the title accordingly

Line 32: add   flooding or flash flood

Added

Line 49: in this session the Authors should add papers dealing with this problem in other contexts of Europe and Italy

We have reviewed and amended the literature review, discussing various other papers in the European and Italian context

Line 104: be more precise about concentration time. Add values.

We have added values and a reference

Lines 281- 284:  this description needs a more detailed figure than figure 6.  I suggest to enlarge the more urbanised are near to the coast, trying to show the main human modification the natural river network.

The figure is now vertical and full page so that the urban area is more visible. In order to provide more details about it, we have added another figure, Figure 9, which is a cartographical comparison of the area in four different periods between 1878 and 1997

194: please try to add more information about the reduction of river width

We have added more information about the reduction of river width and on the increased exposed value in the area following intense urbanisation, population increase and the culverting of the river. This information is supported by the cartographical comparison of Figure 9

402-403: Are you affirming that ”small and widespread interventions should reduce .. time of concentration?  Do you mean increase? Anyway, taking into account rainfall data of Table 3, which show events with very high values of rainfall intensity, mitigation measures have to consider a reduction of the exposed value in association with interventions works in the catchment. The sole change in land use will not be sufficient to risk mitigation.

We have edited the sentence by writing that the time of concentration should be increased. We agree that significant interventions are necessary, some of them already taking place (we refer to those in the Introduction and Discussions, with bibliographical references)

438-442: Is the paper dealing with flooding or with geo-hydrological risk in general ? Please avoid to use this term.

We have replaced the term geo-hydrological risk with flood or flooding hazard

382-390:  The urban expansion cannot be the only reason of the increase of flood frequency in the post war period.  What about climate variations (see lines 379-381)? Urban expansion has increased the exposed value, not the flood frequency. There is a lot of confusion, please delete or re-write these lines

Thanks for this suggestion, we have modified the sentence specifying the increase of exposed value following urbanisation. In the discussions, we also suggest that climate change is something to consider, although we argue that human interference is the main driving factor of the increased flood hazard in the area. 

Tables and Figures

Fig. 1:  add boundaries of sub-catcments listed  in tab 1.; delete admistrative boundaries; ; What does mean the “s “near Bisagno ? stream?; longitude values are not readable. Fig 1b= Are you sure that slope classes are expressed   as degrees? >101°????

We have changed the two figures according to your suggestions. The two figures are now Figure 1 and Figure 2. We decided not to add boundaries of sub-catchments as the small size of the figure would not allow an easy reading of the map and there would be too many information. We think that the use of shaded relief helps with the identification of subcatchments.

Tab 1: please add decimal value to river lenght column.

Added

Fig 2: Please, outline better the river catchment boundary, the outlet and the coastline in this figure and in the following ones (fig. 3 and 4). Add the symbol % in the diagrams

Done

Fig3: add main river network.

Done

Fig. 6: see comment to lines 281-284

Done

Fig. 7: Please add some targets on the photos to allow comparison

Done. We have changed some of the figures to facilitate comparison

 Tab.4: add references for rainfall data, and a column for damages and victims with references.

Done

 Fig.8: Try to express these data as % of the total urban area in each moment.

We have decided not to indicate these data as we think they are not particularly relevant. While in 1822 the urban area was around 0%, in the other flood events it was very near 100%. Instead we added more information about river width, population and exposed value in the area

Round 2

Reviewer 1 Report

Some parts of the manuscript have been improved, but there is still room for improvement.

The research problem remains unclear. Please state it clearly. Literature Review: There is no specific section (different from 1. Introduction and 2. Study area) to review the-state-of-the-art, topic-related scholarly articles and any other sources. This is fundamental to an academic article. Discussions: What are the salient points of the argument? These would make a fine addition to the international scientific literature on flood risk management.

I think the manuscript still needs an improvement, especially focusing on: 1. a clear, straightforward research problem. 2. an individual section of literature review, which is separated from Introduction and Study area. 3. what are the authors' key points of the argument? what are the important things that the authors want to say? This should be new to the field of flood risk management, such as to broaden the definition, a newer type of application and practice...etc.

Author Response

Many thanks for your observations and suggestions; we have made the changes accordingly and we hope that the paper’s aims and key research problem are now clear. We have divided the Introduction into two separate sections: 1 Introduction; 2 Literature Review. The aim of the work, as stated in the Introduction, is a reconstruction of landscape dynamics in the valley at the catchment scale, an approach which differs from previous works on the Bisagno Valley; these are presented in the Literature Review. Given the nature of the special issue (Geomorphological Research for Management and Mitigation of Geo-hyrological Risk and Environmental Sustainability) the work focuses in particular on anthropogenic landforms and geomorphological features. We have added several more relevant references in the Literature Review section. In the Discussion we have added a preliminary paragraph which links with the Introduction section. Considering the results of the paper, we explain why we think that a comprehensive approach to the valley, which also includes rural area is important, an approach which can be used in other similar case studies, particularly in the Mediterranean. We suggest four different strategies to improve flood hazard management in the area in addition to structural works which are already taking place.

Round 3

Reviewer 1 Report

The manuscript has been significantly improved and can be published in the journal.